# Positive Effects of Vitamin D Supplementation in Patients Hospitalized for COVID-19: A Randomized, Double-Blind, Placebo-Controlled Trial

**DOI:** 10.3390/nu14153048

**Published:** 2022-07-26

**Authors:** Sophie De Niet, Mickaël Trémège, Monte Coffiner, Anne-Francoise Rousseau, Doriane Calmes, Anne-Noelle Frix, Fanny Gester, Muriel Delvaux, Anne-Francoise Dive, Elora Guglielmi, Monique Henket, Alicia Staderoli, Didier Maesen, Renaud Louis, Julien Guiot, Etienne Cavalier

**Affiliations:** 1Clinical Department, Laboratoires SMB SA, 1080 Brussels, Belgium; mtrem@smb.be (M.T.); mcoff@smb.be (M.C.); 2Department of Intensive Care, University of Liège, CHU Sart-Tilman, 4000 Liège, Belgium; afrouseau@chuliege.be; 3Department of Pneumology and Allergology, University of Liège, CHU Sart-Tilman, 4000 Liège, Belgium; d.calmes@chuliege.be (D.C.); affrix@chuliege.be (A.-N.F.); f.gester@chuliege.be (F.G.); muriel.delvaux@chuliege.be (M.D.); af.dive@chuliege.be (A.-F.D.); elora.guglielmi@chuliege.be (E.G.); monique.henket@chuliege.be (M.H.); alicia.staderoli@chuliege.be (A.S.); renaud.louis@chuliege.be (R.L.); j.guiot@chuliege.be (J.G.); 4Department of Hospital Pharmacy, CHU Sart-Tilman, 4000 Liège, Belgium; dmaesen@chuliege.be; 5Department of Clinical Chemistry, University of Liège, CIRM, CHU Sart-Tilman, 4000 Liège, Belgium; etienne.cavalier@chu.ulg.ac.be

**Keywords:** vitamin D, cholecalciferol, calcifediol, COVID-19, SARS-CoV-2, hospitalization

## Abstract

Retrospective studies showed a relationship between vitamin D status and COVID-19 severity and mortality, with an inverse relation between SARS-CoV-2 positivity and circulating calcifediol levels. The objective of this pilot study was to investigate the effect of vitamin D supplementation on the length of hospital stay and clinical improvement in patients with vitamin D deficiency hospitalized with COVID-19. The study was randomized, double blind and placebo controlled. A total of 50 subjects were enrolled and received, in addition to the best available COVID therapy, either vitamin D (25,000 IU per day over 4 consecutive days, followed by 25,000 IU per week up to 6 weeks) or placebo. The length of hospital stay decreased significantly in the vitamin D group compared to the placebo group (4 days vs. 8 days; *p* = 0.003). At Day 7, a significantly lower percentage of patients were still hospitalized in the vitamin D group compared to the placebo group (19% vs. 54%; *p* = 0.0161), and none of the patients treated with vitamin D were hospitalized after 21 days compared to 14% of the patients treated with placebo. Vitamin D significantly reduced the duration of supplemental oxygen among the patients who needed it (4 days vs. 7 days in the placebo group; *p* = 0.012) and significantly improved the clinical recovery of the patients, as assessed by the WHO scale (*p* = 0.0048). In conclusion, this study demonstrated that the clinical outcome of COVID-19 patients requiring hospitalization was improved by administration of vitamin D.

## 1. Introduction

The general metabolism and actions of vitamin D in regulating serum calcium concentrations and, in a feedback loop, parathyroid hormone, are well known [1]. There is ample evidence that having enough vitamin D can help prevent many diseases, such as heart disease, bone disease and cancer. Recent data also showed that vitamin D can reduce the risk of respiratory tract infections, and particularly, the risk of viral infections [2,3,4,5,6,7,8,9]. Vitamin D can interfere with viral replication but also has immunomodulatory and anti-inflammatory effects [10,11,12]. These effects can be of importance during severe acute respiratory syndrome coronavirus 2 (SARS-CoV-2) infection. Indeed, SARS-CoV-2 uses immune evasion mechanisms as a common pathogenic mechanism of acute respiratory disease syndrome and systemic inflammatory response syndrome development [13]. Importantly, vitamin D is also involved in renin–angiotensin system regulation. The SARS-Cov-2 virus enters into cells via the angiotensin converting enzyme 2 (ACE 2) receptor, leading to cytokine storms and fatal respiratory distress syndrome [14]. An independent correlation between low serum concentrations of calcifediol (the main vitamin D metabolite) and susceptibility to acute respiratory infection was shown in observational trials [15]. Other studies suggested a protective effect of vitamin D against viral and bacterial respiratory pathogens [3,15,16,17,18]. This positive input of vitamin D is also observed in hospitalized and critically ill patients [19,20]. Indeed, several studies have established a link between a lack of vitamin D and clinical outcomes, such as increased hospital length of stay, readmission rates, sepsis and mortality [19,20,21,22,23,24,25,26,27,28].

Recently, retrospective studies showed a relationship between vitamin D status and coronavirus disease (COVID-19) severity and mortality. The studies observed an inverse relationship between SARS-CoV-2 positivity and calcifediol levels. This correlation persists across latitudes, races, ethnicities, sexes and age ranges [29]. Studies have shown that deficiency in vitamin D was linked to an increased COVID-19 risk and inversely associated with mortality and the need for invasive mechanical ventilation [30,31]. This is supported by the positive results of a double-blind, randomized control trial performed on mechanically ventilated adult intensive care unit (ICU) patients and vitamin D supplementation [32]. Another study recently performed in Spain with patients hospitalized with COVID-19 infection demonstrated that a high dose of calcifediol significantly decreased the need for ICU treatment [33].

Currently available data regarding the beneficial impact of vitamin D in COVID-19 in prospective, interventional and well-designed clinical trials are lacking. There is therefore no consensus regarding vitamin D supplementation for prevention or treatment of COVID-19. Grant et al. suggested that raising serum calcifediol concentrations to 40–60 ng/mL could decrease the risk of COVID-19 infection and death [34]. High prevalence of vitamin D deficiency in patients with comorbidities makes investigations of its role as a secondary therapeutic agent in COVID-19 conceivable. It is therefore necessary to assess serum calcifediol levels in patients with COVID-19 to identify the need for promptly increasing and maintaining the levels of calcifediol in the optimal range. In order to fill this gap, it was of utmost interest to conduct a prospective, interventional study to determine the beneficial effects of vitamin D supplementation as an adjuvant therapy for patients with suboptimal vitamin D status and hospitalized for COVID-19.

The study objective was to assess whether the proposed dosing regimen of a daily dose of 25,000 international units (IU) vitamin D administered over 4 consecutive days, followed by a weekly dose of 25,000 IU, was adequate to rapidly increase the concentrations of calcifediol in patients hospitalized with COVID-19 and to explore its impact on hospital length and other clinical outcomes of the disease.

## 2. Materials and Methods

### 2.1. Methodology

This was an interventional, randomized, parallel, two-treatment, two-arm, double-blind and placebo-controlled pilot study, carried out in one clinical site in Belgium. The study was performed according to the ethical principles of the Declaration of Helsinki, the Good Clinical Practice and the National Drug Law. All patients provided written informed consent, and the study was approved by the Ethics Committee of the University Hospital of Liège, Belgium (local reference: 2020/177, approved on 26 May 2020 and amended on 20 November 2020). The study protocol was registered on ClinicalTrials.gov (Identifier: NCT04636086).

### 2.2. Study Population

All the patients were recruited from the University Hospital of Liège (Liège, Belgium) from August 2020 to August 2021. Caucasian subjects, male and female, aged 18 years or older, with vitamin D deficiency (defined as serum calcifediol concentration ≤ 20 ng/mL) and hospitalized for confirmed SARS-CoV-2 infection at screening were recruited. To be included in the study, the patients were expected to survive for at least 96 h after study entry. The main exclusion criteria were patients presenting acute impairment of renal function or nephrolithiasis. Patients with hypercalcemia and/or hypercalciuria and/or pseudohypoparathyroidism were also excluded at screening. Concomitant medications susceptible to interfere with the study results were not allowed, and subjects who had used any type of vitamin D supplement at screening visit were excluded.

### 2.3. Study Intervention

The patients were randomized in the two different treatment groups (vitamin D or placebo) in a 1:1 ratio. Patients participated in the study for a maximum of 9 weeks, including an up to 6-week treatment period and a maximum of 3-week follow-up period. The study duration was defined by the length of patient’s hospitalization. The patients stayed at the hospital during the overall treatment period. The last day of treatment period was the last day of hospitalization or day 36, whichever came first. The intervention group received the best available treatment plus oral vitamin D (ampoule of 1 mL containing 25,000 IU of cholecalciferol (vitamin D_3_), D-CURE^®,^ Laboratoires SMB SA). A daily dose of 25,000 IU of vitamin D over 4 consecutive days was given to rapidly restore the calcifediol levels. Then, 25,000 IU per week up to six weeks was given to maintain this level. This dose was proposed on the basis of previous studies performed with oral vitamin D [35,36,37]. The control group received the best available treatment for COVID plus placebo (ampoule of 1 mL of excipient). The study medications were identical in consistency, smell and taste. The placebo and active study treatments were given either orally or via enteral feeding tube. The study was double blinded. The patients, investigators and any other persons involved in the data handling were blinded to the trial medications. To maintain the blind, the calcifediol levels were not provided to the clinical staff after randomization until database lock. The patients in ICU with enteral nutrition received 600 IU vitamin D per day in addition to the study treatments, assuring a standard supplementation of vitamin D to patients with possibly more severe deficiency in vitamin D. The study design is summarized in Table 1.

### 2.4. Study Endpoints

Outcomes of effectiveness included calcifediol serum level, ordinal scale for clinical improvement as recommended by the World Health Organization (WHO), hospitalization length, intensive care unit admission, time until absence of fever, need for supplemental oxygen, non-invasive ventilation, high-flow oxygen devices, invasive mechanical ventilation or additional organ support and death [38].

### 2.5. Laboratory Tests

Clinical specimens required for SARS-CoV-2 diagnostic procedures were collected on admission by nasopharyngeal exudate sampling following WHO guidelines and recommendations [39]. Blood samples were collected to determine the serum concentration of calcifediol. The Fujirebio 25-OH Vitamin D assay on Lumipulse G1200 analyzer (Fujirebio, Tokyo, Japan) was used to screen the calcifediol concentrations of the patients at the inclusion of the study. This assay indeed showed excellent concordance with the LC–MS/MS method used in our laboratory and has the advantage of providing results in a fast turnaround time, compatible with the needs of a screening [40,41,42]. All the other samples were measured in a single batch with our LC–MS/MS method, which is certified by the Vitamin D Standardization and Certification Program (VDSCP) to be traceable to the Centers for Disease Control reference method. The details of our methods have been published previously [43]. All analyses were performed at the ISO 15,198 clinical chemistry laboratory of the University of Liège (Liège, Belgium).

### 2.6. Statistical Analyses

The number of participants was determined on the basis of feasibility, based on resources, capacity of clinical staff and available patients. Given that the minimal clinically important difference between the groups for length of stay among patients with COVID-19 is unknown, no formal calculation of sample size was performed, and it was decided that 25 subjects per treatment group would be included. All statistical computations were performed using the SAS/STAT software version 9.4. (SAS Institute Inc., Cary, NC, USA). Continuous variables were presented as mean ± standard deviation (median and interquartile range for non-normalized data), and categorical variables were presented as frequencies and percentages. The endpoints were compared between treatment groups and were analyzed as follows: categorical variables were analyzed using the chi-square test or Fisher’s exact test for categorical variables and independent-sample *t*-test for continuous variables. Furthermore, in case the Student’s *t*-test was not applicable for parameters not normally distributed, a non-parametric Wilcoxon (Mann–Whitney U-test) was applied to compare the two independent groups (placebo vs. Vitamin D) for these parameters. A 2-sided *p* value of less than 0.05 was considered significant. A post hoc adjusted analysis for the outcome length of hospital stay was performed by estimating a multiple regression equation relating the outcome of interest to independent variables representing the treatment assignment—the co-founding variables. These cofounders were age, gender, height, weight, BMI, arterial hypertension, diabetes, hepatic failure, renal failure, cardiac pathology, chronic lung disease and vaccinal status. The number of cofounders being high with regard to the sample size, two different models were established.

## 3. Results

### 3.1. Baseline Characteristics

A total of 50 subjects signed their informed consent and were randomized to either the vitamin D group (*n* = 26) or the placebo group (*n* = 24). A total of 43 of them completed the study: 21 in the vitamin D group and 22 in the placebo group (Figure 1). Out of the 26 patients randomized in the vitamin D group, 4 of them withdrew from the study or were discharged from hospital before the planned treatment administration, and 1 of them refused to take any study treatments. Out of the 24 patients randomized in the placebo group, 1 of them was discharged from the hospital before the planned study treatment administration, and 1 refused to take any study treatments. Baseline demographic and clinical characteristics were similar in both groups (*p* > 0.05) (Table 2). At baseline, comorbidities, including cardiac and lung diseases, renal and hepatic failure, diabetes, arterial hypertension and body mass index (BMI), did not differ between groups. The vaccinal status was comparable across both groups, the majority of the subjects not being vaccinated.

### 3.2. Safety Assessment

The proposed dosing regimen was well tolerated, and no specific adverse events in relation to vitamin D supplementation were identified during the study.

### 3.3. Measurements of Calcifediol

Baseline mean serum calcifediol concentrations were comparable across the two groups (*p* = 0.7415) and below 20 ng/mL (Table 2). After supplementation, in the vitamin D group, the calcifediol concentrations rapidly increased, reaching mean value of 29.9 ± 14.81 ng/mL, whereas no changes vs. baseline were observed in the placebo group.

### 3.4. Clinical Outcomes

The median length of hospital stay significantly decreased in the vitamin D group compared to the placebo group (4 days for the vitamin D group vs. 8 days for the placebo group; *p* = 0.003) (Table 3). These results were confirmed by the adjusted post hoc analysis (Unadjusted treatment effect size for hospital length of stay −7.22 [−13.16; −1.29], *p* = 0.0183; Adjusted treatment effect size −7.35 [−14.10; −0.59], *p* = 0.034 and −8.76 [−15.88; −1.64], *p* = 0.018). At Day 7, a significantly lower proportion of patients were still hospitalized in the vitamin D group than in the placebo group (19% vs. 54%; *p* = 0.0161), and none of the patients treated with vitamin D were hospitalized after 21 days compared to 14% of the patients treated with placebo (Figure 2). Multiple regression models were performed to analyze the relationship between the primary endpoint “hospital length of stay” and the identified cofounders. The analysis confirmed there was no effect of age, BMI, height, weight, gender, cardiac pathology, arterial hypertension, diabetes, hepatic failure and vaccinal status on the primary endpoint (*p* > 0.05). The relationship was independent, regardless of the treatment administered (Interaction treatment*cofounders *p* > 0.05). Five patients were admitted to the intensive care unit in the placebo group vs. two in the vitamin D group (23% vs. 9.5%; *p* = 0.4121). The average intensive care unit length of stay was shorter in the vitamin D group compared to the placebo group (4.0 days ± 4.2 vs. 12.4 days ± 14.3; *p* = 0.4724). A positive trend of supplementation with vitamin D was also noticed in a smaller proportion of patients who needed supplemental conventional oxygen, non-invasive ventilation, high-flow nasal oxygen and invasive mechanical ventilation (86.4% in the placebo group vs. 62% in the vitamin D group; *p* = 0.0545). Among all the patients who needed supplemental conventional oxygen, the administration of vitamin D significantly decreased the duration of treatment (4 days vs. 7 days; *p* = 0.012). The duration of recovery from fever was also shorter in the vitamin D group but without statistical significance (7.7 days ± 4.7 vs. 14.1 days ± 13.1; *p* = 0.0593). No significant differences were observed regarding mortality. Three patients died due to COVID-19 in the placebo group and one in the vitamin D group (Table 3).

### 3.5. WHO Scale

At randomization, the severity of the disease assessed by the ordinal WHO scale for clinical improvement was comparable across both groups, the majority of the patients (95%) being categorized in the moderate clinical assessment. After vitamin D supplementation, a more rapid and greater improvement was observed compared to the placebo group (Table 4). At Day 7, 71% of the patients supplemented with vitamin D switched from the moderate to the mild category of the scale compared to 18% in the placebo group (*p* = 0.0048). At day 36, 90% of the patients from the vitamin D group were assessed as “no more infected” compared to 77% in the placebo group.

## 4. Discussion

In this randomized, double-blind, placebo-controlled clinical trial, a proposed dosing regimen of 25,000 IU vitamin D per day over four consecutive days followed by 25,000 IU per week did significantly reduce the hospitalization length, the need for supplemental oxygen, and it improved the WHO score among patients with COVID-19. In our trial, all efforts were made to adequately control the possible cofounding factors. All patients should have a calcifediol level ≤ 20 ng/mL at study entry to have comparable baseline mean values. The study treatment was taken under the supervision of the clinical staff, which led to 100% compliance in both treatment groups. All patients were recruited from one site. Therefore, the standard of care treatment and the management of patients were the same for all patients. It is interesting to highlight that both groups of patients had comparable percentage of unfavorable risk factors at baseline. Indeed, there was no significant difference in subjects regarding age, sex, BMI, diabetes, arterial hypertension and cardiac, hepatic, renal or lung disorders. Moreover, the multiple regression analysis confirmed these cofounding factors could be ruled out as possible bias. It should also be noted that the vaccinal status was comparable for both groups, the majority of the patients being not yet vaccinated. Importantly, all the patients received the same best available treatment at the hospital, which included the use of corticosteroids, paracetamol, heparin and remdesevir.

It was observed that higher doses of vitamin D than usual were given in case of deficiency in patients at risk of developing respiratory infections [34]. Our study assessed a dosing regimen of 25,000 IU vitamin D per day for four consecutive days, followed by 25,000 IU per week until discharge. The study results confirmed that this regimen was adequate to rapidly raise the calcifediol level above 20 ng/mL and improve the clinical outcome of patients requiring hospitalization for COVID-19. Whether that would also apply to patients with an earlier stage of the disease is unknown.

Our study complements the recent findings of Castillo et al., 2020 study performed in Spain with 76 patients hospitalized with COVID-19 infection, which showed that the administration of high dose of calcifediol (21,280 IU on the day of admission, 10,640 IU on day 3 and 7, and then weekly until discharge) significantly reduced the need for ICU treatment of patients requiring hospitalization due to proven COVID-19 (*p <* 0.001) [33].

This is also supported by the positive results of a double-blind, randomized control trial performed on mechanically ventilated adult ICU patients and vitamin D supplementation in two hospitals in Atlanta [32]. Subjects were administered either placebo, 50,000 IU vitamin D or 100,000 IU vitamin D daily for 5 consecutive days enterally. The high-dose vitamin D safely increased plasma calcifediol concentrations to the sufficient range and was associated with decreased hospital length of stay (25 ± 14 and 18 ± 11 days compared to 36 ± 19 days in placebo group; *p* = 0.03).

The lack of benefit observed in the Brazilian study of Murai et al., 2021 could have been affected by the single higher dose of vitamin D administered once as a bolus dose (200,000 IU) and the heterogeneity of the sample underlined by the authors due to several coexisting diseases and diverse medication regimens [44]. The same conclusions were drawn in the recent COVID-VIT-D trial where a single oral bolus of 100,000 IU vitamin D did not improve the outcomes of the disease compared with patients who did not receive it, even if the cohort analysis showed that high serum calcifediol level at hospital admission was associated with better outcomes [45]. As observed by the authors, the different sun exposure of the patients being recruited from different countries with different latitudes could have influenced the calcifediol levels and the study results. In our study, all patients were recruited over one year and could have been potentially impacted by the seasonal changes [46,47]. However, they were recruited from the same hospital and consecutively assigned to the active or the placebo group considering a 1:1 ratio, decreasing the risk of having unbalanced randomization in terms of any potential seasonal changes. Moreover, all patients stayed at the hospital during the whole study period, which avoided any bias related to sun exposure. Therefore, the potential impact of the seasonal changes on the human immune response can be ruled out.

Our study adds to the current literature of vitamin D supplementation as an adjuvant therapy for patients with COVID-19 requiring hospitalization, particularly by focusing on patients with vitamin D deficiency (≤20 ng/mL).

The strengths of the study include the randomized, placebo-controlled and double-blind experimental design, the comparable groups at baseline and the assessment of calcifediol serum level with clinical outcomes at baseline and during the study. The study shows some limitations, which are the low number of subjects and the minimal clinically important difference in hospital length among patients with COVID-19, which remains to be determined. Our study showed that vitamin D deficiency is an easily modifiable risk factor for COVID-19 that should be actively corrected. Vitamin D supplementation is a simple, safe and inexpensive measure, which is effective in correcting hypovitaminosis D. Even a small improvement in COVID-19 clinical outcome would easily justify this intervention.

## 5. Conclusions

In this study, including COVID-19 patients requiring hospitalizations, administration of cholecalciferol significantly reduced the hospital length of stay, reduced the duration of supplemental oxygen and improved the clinical status assessed by the WHO scale. Further studies with a larger number of patients would be needed to confirm our observations.

## Figures and Tables

**Figure 1 nutrients-14-03048-f001:**
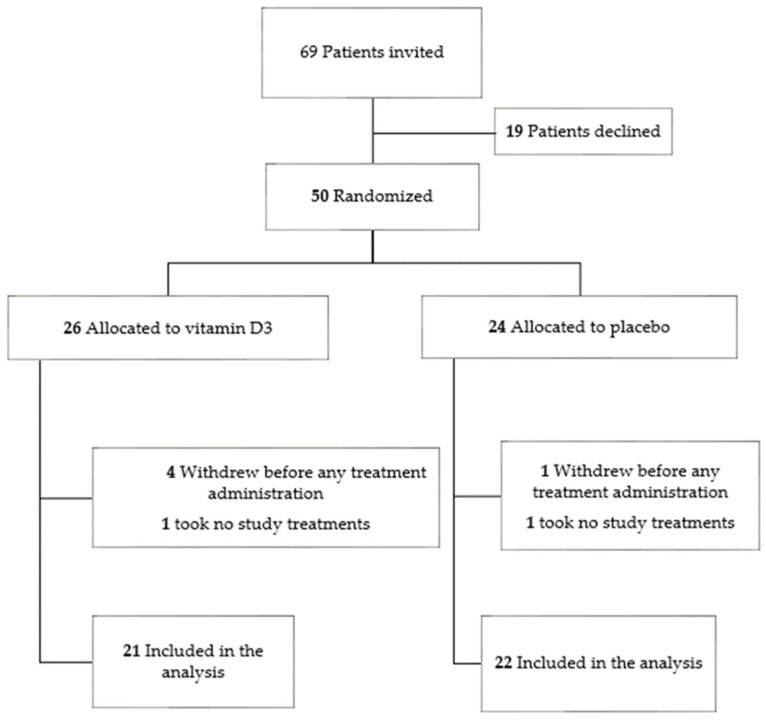
Flow chart of patients included in the study.

**Figure 2 nutrients-14-03048-f002:**
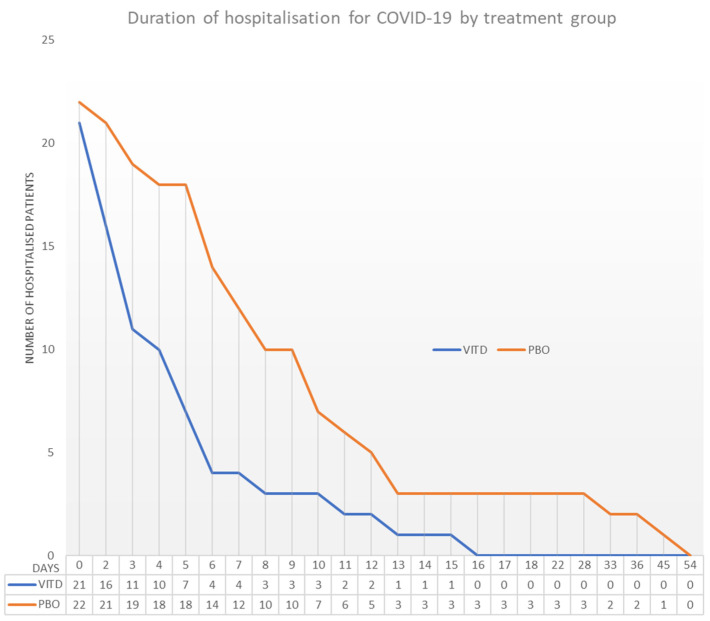
Number of hospitalized patients by treatment group.

**Table 1 nutrients-14-03048-t001:** Study design.

	Day 1, 2, 3, 4, 8, 15, 22, 29 and 36
Group 1: Placebo group	1 ampoule of placebo + standard of care treatment
Group 2: Vitamin D group	1 ampoule of vitamin D 25,000 IU + standard of care treatment

**Table 2 nutrients-14-03048-t002:** Baseline demographic characteristics.

		Placebo Group *n* = 22	Vitamin D Group *n* = 21	*p* Value
Age (years)				
	Mean ± SD	68.73 ± 10.97	63.24 ± 14.46	0.167
	min–max	41.00–88.00	36.00–98.00	
Gender				
Male	*n* (%)	10 (45%)	13 (62%)	0.364
Female	*n* (%)	12 (54%)	8 (38%)	
Weight (kg)				
	Mean ± SD	79.46 ± 17.54	75.85 ± 12.03	0.728
	min–max	47.70–106.00	63.00–108.00	
BMI (kg/m^2^)				
	Mean ± SD	28.92 ± 6.92	26.52 ± 3.24	0.133
	min–max	19.11–44.89	21.30–32.97	
Calcifediol concentration at screening (ng/mL)	*n*	22	21	
	Mean ± SD	16.87 ± 9.48	17.87 ± 10.15	0.741
	min–max	4.80–43.60	5.00–44.60	
Vaccinal Status				
Not vaccinated	*n* (%)	20 (91%)	17 (81%)	0.169
1 Dose	*n* (%)	2 (9.1%)	1 (5%)	
2 Doses	*n* (%)	0 (0.0%)	3 (14%)	
Cardiac Pathology				
Yes	*n* (%)	9 (42%)	7 (33%)	
Hepatic Failure				
Yes	*n* (%)	0 (0.0%)	2 (9.5%)	
Renal Failure				
Yes	*n* (%)	4 (18%)	4 (19%)	
Diabetes				
Yes	*n* (%)	8 (36%)	8 (38%)	
Arterial Hypertension				
Yes	*n* (%)	13 (59%)	11 (52%)	
Chronic lung disease				
Yes	*n* (%)	9 (42%)	5 (24%)	

BMI: body mass index; SD: standard deviation.

**Table 3 nutrients-14-03048-t003:** Clinical outcomes.

		Placebo Group *n* = 22	Vitamin D Group *n* = 21	*p* Value
Hospital length of stay (Days)				
	Median	8.0	4.0	0.003
	Q1–Q3	6.0–12.0	3.0–6.0	
Proportion of patients hospitalized				
At Day 7	*n* (%)	12 (54)	4 (19.)	0.016
At Day 14	*n* (%)	3 (14)	1 (4.8))	0.262
At Day 21	*n* (%)	3 (14)	0 (0.0)	0.125
At Day 28	*n* (%)	3 (14)	0 (0.0)	0.125
At Day 36	*n* (%)	2 (9.1%)	0 (0.0)	0.256
Admission in intensive care unit	*n* (%)	5 (23)	2 (9.5)	0.412
Intensive Care Unit length of stay (Days)	Mean ± SD	12.4 ± 14.3	4.0 ± 4.2	0.472
	min–max	3.0–36.0	1.0–7.0	
Proportion of patients requiring supplemental oxygen, non-invasive ventilation or high-flow oxygen devices, invasive mechanical ventilation	*n* (%)	19 (86)	13 (62)	0.054
Duration of supplemental conventional oxygen (Days)				
	Median	7.0	4.0	0.012
	Q1–Q3	5.0–11.0	0.0–6.0	
Duration of non-invasive ventilation or high-flow nasal oxygen, invasive mechanical ventilation or additional organ support (Days)	Mean ± SD	1.3 ± 4.2	0.3 ± 1.3	0.306
	min–max	0.0–16.0	0.0–16.0	
Time until absence of fever for more than 48 h without antipyretics (Days)	Mean ± SD	14.1 ± 13.1	7.7 ± 4.7	0.059
	min–max	0.0–52.0	2.0–18.0	
Mortality All causes	*n* (%)	3 (14)	4 (19)	0.286
Mortality related to COVID-19	*n* (%)	3 (12)	1 (4.8)	0.129

**Table 4 nutrients-14-03048-t004:** WHO ordinal scale.

		Placebo Group *n* = 22	Vitamin D Group *n* = 21	*p* Value
Ordinal Scale for clinical improvement by severity * at baseline (Day 1)				0.512
No infection	*n* (%)	0 (0.0)	0 (0.0)	
Mild	*n* (%)	0 (0.0)	0 (0.0)	
Moderate	*n* (%)	21 (95)	20 (95)	
Severe	*n* (%)	1 (4.5)	1 (4.8)	
Death	*n* (%)	0 (0.0)	0 (0.0)	
Ordinal Scale for clinical improvement by severity * at Day 7				0.005
No infection	*n* (%)	0 (0.0)	0 (0.0)	
Mild	*n* (%)	4 (18)	15 (71)	
Moderate	*n* (%)	14 (63)	4 (19)	
Severe	*n* (%)	3 (13)	1 (4.8)	
Death	*n* (%)	1 (4.5)	1 (4.8)	
Ordinal Scale for clinical improvement by severity * at Day 15				0.549
No infection	*n* (%)	0 (0.0)	0 (0.0)	
Mild	*n* (%)	15 (68)	18 (85)	
Moderate	*n* (%)	4 (18)	1 (4.8)	
Severe	*n* (%)	1 (4.5)	1 (4.8)	
Death	*n* (%)	1 (4.5)	1 (4.8)	
Ordinal Scale for clinical improvement by severity * at Day 22				0.543
No infection	*n* (%)	0 (0.0)	0 (0.0)	
Mild	*n* (%)	17 (77)	19 (90)	
Moderate	*n* (%)	1 (4.5)	0 (0.0)	
Severe	*n* (%)	2 (9.1)	0 (0.0)	
Death	*n* (%)	2 (9.1)	2 (9.5)	
Ordinal Scale for clinical improvement by severity * at Day 29				0.543
No infection	*n* (%)	0 (0.0)	0 (0.0)	
Mild	*n* (%)	17 (77)	19 (90)	
Moderate	*n* (%)	1 (4.5)	0 (0.0)	
Severe	*n* (%)	2 (9.1)	0 (0.0)	
Death	*n* (%)	2 (9.1)	2 (9.5)	
Ordinal Scale for clinical improvement by severity * at Day 36				0.318
No infection	*n* (%)	0 (0.0)	0 (0.0)	
Mild	*n* (%)	17 (77.)	19 (90)	
Moderate	*n* (%)	2 (9.1)	0 (0.0)	
Severe	*n* (%)	0 (0.0)	0 (0.0)	
Death	*n* (%)	3 (14)	2 (9.5)	

* No infection = 0; Mild = Score 1, 2; Moderate = Score 3, 4, Severe = Score 5, 6, 7 and Death = Score 8.

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
