# Peer review of "Positive Effects of Vitamin D Supplementation in Patients Hospitalized for COVID-19: A Randomized, Double-Blind, Placebo-Controlled Trial"

_nutrients, 2022, doi:10.3390/nu14153048_

Round 1
Reviewer 1 Report
The study deals with an important topic. There are some major points that must be addressed. Please consider the following criticisms:
Length of hospital stay is likely not normally distributed. Therefore, it should not be reported as media and SD, but as median and interquartile range. Analyses should be also changed accordingly. The same is for the duration of supplemental oxygen.
The randomization strategy is not reported
“A daily dose of 25.000 IU of vitamin D over 4 consecutive days was given to rapidly restore 123 the calcifediol levels.” This sentence is tricky since the authors state that the study is double blind. However, if the researchers found an increase of vitamin D levels this implies that patients received vitamin d and not placebo and the blindness could not be guaranteed
No power analysis is performed: the manuscript should be reported as preliminary/pilot study
The analyses are too superficial. There is a large number of confounders that should be considered. At least the authors should perform models which include the age of patients, the sex, the BMI, vaccinal status/previous SARS-CoV-2 infection and presence of significant underlying commorbidities. I understand that the authors report such variables in table 1, but their potential effect should be tested in multiple regression models.
Recent studies also found a potential seasonal dependence of vitamin D effect (PMID: 33420273, PMID: 25965853). These articles should be discussed. Furthermore, since the study was performed over one year, the season should be considered as adjustment in multiple regression models. This would add much interesting information
Analyses should be performed not only for protocol but also for intention to treat
How many missing data were present?
How were they managed?
The patient flowchart should also include the number of invited patients and the number of declining patients
Minor comments
I suggest discussing more in general data on vitamin D and pandemic (e.g., PMID: 33925932)
Percentages equal or >10 should be reported without decimals. Overall, all variables should be reported with significant digits (e.g. “days” should be reported with one decimal)
Authors should describe the efforts employed to avoid all bias
Author Response
We appreciate the reviewer’s comments. The followings are our point-by-point responses:
- Reviewer 1- Comment 1: Length of hospital stay is likely not normally distributed. Therefore, it should not be reported as media and SD, but as median and interquartile range. Analyses should be also changed accordingly.
Reviewer 1- Comment 2: The same is for the duration of supplemental oxygen.
Answer 1 and 2: As suggested by the reviewer, the length of hospital stay and the duration of supplemental oxygen are presented as median and interquartile range. Furthermore, the non-parametric Wilcoxon test (Mann Whitney U test) was applied for the comparison placebo versus Vitamin D. The table 2 of the manuscript was modified with the below data.
|
|
Placebo Group N = 22 |
Vitamin D group N = 21 |
P Value |
Hospital length of stay (Days) |
|
|
|
|
|
Median |
8.0 |
4.0 |
0.003 |
|
Q1-Q3 |
6.0-12.0 |
3.0-6.0 |
|
|
Interquartile range |
6.0 |
3.0 |
|
Duration of supplemental conventional oxygen (Days) |
|
|
|
|
|
Median |
7.0 |
4.0 |
0.012 |
|
Q1-Q3 |
5.0-11.0 |
0.0-6.0 |
|
|
Interquartile range |
6.0 |
6.0 |
|
- Reviewer 1- Comment 3: The randomization strategy is not reported. “A daily dose of 25.000 IU of vitamin D over 4 consecutive days was given to rapidly restore the calcifediol levels.” This sentence is tricky since the authors state that the study is double blind. However, if the researchers found an increase of vitamin D levels this implies that patients received vitamin d and not placebo and the blindness could not be guaranteed.
Answer 3: After confirmation that the patient met all eligibility criteria for the study, the patients were randomized and assigned a randomization number allocated by chronological order. The patients were assigned at random to one of the 2 treatment groups (active or placebo) according a randomization list considering a 1:1 ratio between active and placebo treatment. This study was double-blinded. The patients, investigator(s) and any other persons involved in the monitoring, the data handling or the conduct of the trial were blinded to the trial medication. The packaging and labelling of study medications were carried out by an independent person. To maintain the blind, the ampoules of cholecalciferol were visually match the ampoules containing the placebo. At screening, the assessment of vitamin D level was done by Fujirebio 25-OH Vitamin D assay on Lumipulse G1200 analyzer. This assay has indeed the advantage of providing results in a fast turnaround time, compatible with the needs of a screening. After randomization, to maintain the blind, the results of vitamin D levels were no more provided to the clinical site until database lock. Indeed, all the other samples taken during the study were measured in a single batch and measured with a LC–MS/MS method at the end of the study. The maintain of the blind was described in more details in the manuscript in the section “2.3 Study intervention”.
- Reviewer 1- Comment 4: No power analysis is performed: the manuscript should be reported as preliminary/pilot study.
Answer 4: The term “pilot” study was added in the section “2.1 Methodology” of the manuscript.
- Reviewer 1- Comment 5: The analyses are too superficial. There is a large number of confounders that should be considered. At least the authors should perform models which include the age of patients, the sex, the BMI, vaccinal status/previous SARS-CoV-2 infection and presence of significant underlying comorbidities. I understand that the authors report such variables in table 1, but their potential effect should be tested in multiple regression models.
Answer 5: As requested by the reviewer, multiple regression models were performed to analyze the relationship between the primary endpoint “hospital length of stay” and the identified cofounders. The results, presented in the table below, confirmed there is no effect of age, BMI, height, weight, gender, cardiac pathology, arterial hypertension, diabetes, hepatic failure and vaccinal status on the primary endpoint (p>0.05). The relationship is independent regardless the treatment administered (Interaction treatment*cofounders p>0.05).
Cofounders |
Pr > F |
Age |
0.3758 |
Age*Treatment |
0.8283 |
BMI |
0.7244 |
BMI*Treatment |
0.8581 |
Height |
0.2890 |
Height*Treatment |
0.9758 |
weight |
0.6486 |
weight*Treatment |
0.8816 |
Gender |
0.6695 |
Gender*Treatment |
0.4625 |
Cardiac Pathology |
0.3352 |
Cardiac Pathology *Treatment |
0.3417 |
Arterial Hypertension |
0.3804 |
Arterial Hypertension *Treatment |
0.8542 |
Diabete |
0.4116 |
Diabete *Treatment |
0.2306 |
Hepatic Failure |
0.4354 |
Hepatic Failure *Treatment |
0.6651 |
- Reviewer 1 – Comment 6: Recent studies also found a potential seasonal dependence of vitamin D effect (PMID: 33420273, PMID: 25965853). These articles should be discussed. Furthermore, since the study was performed over one year, the season should be considered as adjustment in multiple regression models. This would add much interesting information
Answer 6: The author thanks the reviewer for this interesting comment regarding the seasonal impact. The recommended articles were deeply reviewed. Dopico et al reported a variability in the immune system according to seasonal period, which could increase the differences in some immune responses between subjects if blood samples were collected at different times of year. In the study, the seasonal periods were classified as winter from December to February and summer from June to August. To assess if there was a potential impact of the seasonal period in our study, we defined four seasonal periods classified as: winter from December to February, spring from March to May, summer from June to August and autumn from September to November. The recruitment of the study sorted by these four seasonal periods is presented in the table below.
Treatment Group |
Season |
|||
Autumn |
Spring |
Summer |
Winter |
|
PBO (N=22) |
2 |
11 |
2 |
7 |
VITD (N=21) |
4 |
9 |
2 |
6 |
Overall, the data presented in this table showed that the recruitment of the patients was well balanced between the two treatment groups in terms of seasonal period, that decreased the risk of potential bias related to the seasonal changes. The discussion about the potential seasonal impact was added in the publication as proposed: “In our study, all patients were recruited over one year and could have been potentially impacted by the seasonal changes [46,47]. However, they were recruited from the same hospital and consecutively assigned to the active or the placebo group considering a 1:1 ratio; decreasing the risk to have unbalanced randomization in terms of any potential seasonal changes. Moreover, all patients stayed at the hospital during the whole study period which avoided any bias related to the sun exposure. Therefore, the potential impact of the seasonal changes on the human immune response can be ruled out. “
- Reviewer 1- Comment 7: Analyses should be performed not only for protocol but also for intention to treat
Answer 7: The statistical analysis was performed using the ITT set including 21 patients in the vitamin D group and 22 patients in the placebo group. Considering that no major deviations were reported in this study by any of the patients who took at last one dose of the study treatments, the Per Protocol (PP) population was the same than the ITT set and therefore no additional analyses was performed. Indeed, a total of 50 subjects signed their informed consent and were randomized to either the vitamin D group (N=26) or the placebo group (N=24). Forty-three (43) of them completed the study, 21 in the vitamin D group and 22 in the placebo group. Over the 26 patients randomised in the vitamin D group, 4 of them withdrew from the study or were discharged from hospital before the planned treatment administration and 1 of them refused to take any study treatments. Over the 24 patients randomised in the placebo group, one of them was discharged from the hospital before the planned study treatment administration and another refused to take any study treatments. The intention-to-treat approach, which provides a conservative strategy, implies that the analysis should be performed including all randomised subjects. Among all the randomised patients, the patients who were discharged before the planned study treatments administration were excluded from the ITT analysis set as no post randomisation data were available. The two patients who took no study treatments (1 in each study arm) during the whole study period were excluded as well from the ITT population as recommended in the ICHE9 (CPMP/ICH/363/96) “In some situations, it may be reasonable to eliminate from the set of all randomised subjects any subject who took no trial medication”. The study being performed in double bling fashion, the decision of these two patients to not take the study treatment was not leaded by knowledge of the assigned treatment and therefore no bias could be identified. The figure 2 presented in the publication was updated to clarify the analysis set used for the statistical analysis.
- Reviewer 1- Comment 8: How many missing data were present?
Reviewer 1- Comment 9: How were they managed?
Answer 8 and 9: No missing data were reported for the 43 patients included in the analysis. Regarding the WHO Ordinal Scale for clinical improvement endpoint, the last available score assessment was carried forward for all the timepoints planned after the patient’s discharge (LOCF method).
- Reviewer 1 – Comments 10: The patient flowchart should also include the number of invited patients and the number of declining patients
Answer 10: The number of invited and declined patients was added to the patient flow chart.
- Reviewer 1- Comment 11: I suggest discussing more in general data on vitamin D and pandemic (e.g., PMID: 33925932).
Answer 11: The PMID 33925932 article discussed the vitamin D status of adolescent during the COVID-19 pandemic and showed no higher risk of vitamin D insufficiency. Our study focused on adult hospitalized patients with vitamin D insufficiency. In our study, the mean age of the included patients was above 60 years. It is therefore no clear for the author how a link should be done with the PMID 33925932 article. Could the reviewer clarify her/his comment?
- Reviewer 1 – Comment 12: Percentages equal or >10 should be reported without decimals. Overall, all variables should be reported with significant digits (e.g. “days” should be reported with one decimal).
Answer 12: The decimals were modified as suggested.
- Reviewer 1 – Comment 13: Authors should describe the efforts employed to avoid all bias.
Answer 13: In our trial, all efforts were done to adequately control the confounding factors.
- All patients should have a calcifediol level ≤ 20 ng/ml at study entry.
- All patients were hospitalized from randomization until the end of the treatment period. Therefore, vitamin D uptake from sun exposure was avoided and intake of vitamin D from food was standardized for all patients.
- The study treatment was taken under the supervision of the clinical staff which led to a 100% compliance in both treatment groups.
- All patients were recruited from one site. Therefore, the standard of care treatment and the management of patients were the same for all patients.
- The age of the patients, the sex, the BMI, the vaccinal status/previous SARS-CoV-2 infection and the presence of significant underlying comorbidities were recorded to confirm the comparability of the groups. The adjusted analysis confirmed these factors can be ruled out as possible bias.
- All patients were recruited from the same hospital and consecutively assigned to the active or the placebo group considering a 1:1 ratio; leading to well-balanced groups in terms of any potential seasonal changes on the human response.
The description of these confounding factors was added in the discussion section of the publication.
Reviewer 2 Report
The work employs both scientific and practical reasoning. The paper is moreover focused on investigating and describing the linkage between rapidly supplementation of vitamin D (25.000 IU) on serum concentrations in patients hospitalized with SARS-CoV-2 infection and to explore its impact on clinical outcomes of the disease. The methodology is modern and adequate to the subject, however might require a further comment to make the reasoning more explicit (2.5. Laboratory Tests). Modern methods of statistical analysis allow good execution of scientific content. The results are correctly presented and referred to in comments. The summary is concise and adherent to the subject. Bibliography is adequate and consistent.
Author Response
We appreciate the reviewer’s comments. The followings are our point-by-point responses:
Reviewer 2 – Comment 1: The methodology is modern and adequate to the subject, however might require a further comment to make the reasoning more explicit (2.5. Laboratory Tests).
Answer 1: The section 2.5 (Laboratory tests) was modified in order to make the reasoning more explicit. The updated section is below with tracked changes.
“Clinical specimens required for SARS-CoV-2 diagnostic procedures were obtained on admission by nasopharyngeal exudate sampling following WHO guidelines and recommendations[39]. Blood samples were collected to determine the serum concentration of calcifediol. The Fujirebio 25-OH Vitamin D assay on Lumipulse G1200 analyzer (Fujirebio, Tokyo, Japan) was used to screen the calcifediol concentrations of the patients at the inclusion of the study. This assay has indeed shown excellent concordance with the LC-MS/MS method used in our laboratory and has the advantage of providing results in a fast turnaround time, compatible with the needs of a screening [40–42]. All the other samples were measured in a single batch with our LC–MS/MS method, which is certified by the Vitamin D Standardization and Certification Program (VDSCP) program to be traceable to the Centers of Diseases Control reference method. The details of our methods have been published previously [43]. All the analyses were performed at the ISO 15198 clinical chemistry laboratory of the University of Liège (Liège, Belgium).”
Round 2
Reviewer 1 Report
I thank the authors for their efforts to revise the manuscript. The authors addressed most of my comments. The paper has now improved.
I have still two remarks:
- confounders in the multiple regressions analyses should be used to verify if the effect of vitamin D administration is confirmed even after adjustments. However, I could not find this. Please report a table showing that the effect is present also after adjustments. this would markedly increase the quality of the results. If the number of confounders is too high considering the sample size, the authors might perform two different models.
- I think that the term "pilot study" should be introduced also in the title and/or in the abstract.
Author Response
We appreciate the reviewer’s comments. The followings are our point-by-point responses:
Reviewer 1-Round 2- Comment 1: Confounders in the multiple regression’s analyses should be used to verify if the effect of vitamin D administration is confirmed even after adjustments. However, I could not find this. Please report a table showing that the effect is present also after adjustments. this would markedly increase the quality of the results. If the number of confounders is too high considering the sample size, the authors might perform two different models.
Answer 1: Multiple regression analysis can be indeed used to assess effect modification from confounders. This is done by estimating a multiple regression equation relating the outcome of interest (hospital length of stay) to independent variables representing the treatment assignment, confounding variables and the product of the two (called the treatment by cofounding interaction variable). The fact that the interaction variables are not statistically significant indicates that the association between treatment and hospital length of stay is not affected by any of the identified co-founders. Further to the comment of the reviewer, the adjusted treatment effect was assessed for the outcome length of hospital stay with a post-hoc adjusted analysis by estimating a multiple regression equation relating the outcome of interest (hospital length of stay) to independent variables representing the treatment assignment, co-founding variables. These cofounders were age, gender, height, weight, BMI, arterial hypertension, diabetes, hepatic failure, renal failure, cardiac pathology, chronic lung disease and vaccinal status. The numbers of cofounders being high in regards with the sample size, two different models were considered to determine the adjusted treatment effect. The following confounders were considered in the model 1: age, gender, height, weight and BMI and the model 2 was built using the arterial hypertension, diabetes, hepatic failure, renal failure, cardiac pathology, chronic lung disease and vaccinal status as cofounders. The adjusted post-hoc analysis confirmed the effect of vitamin D administration over the hospital length of stay as shown is the following table.
Model |
Treatment effect (Contrast) |
95% Confidence interval |
p-value |
Unadjusted or crude model |
-7.22 |
[-13.16; -1.29] |
0.0183 |
Adjusted model 1 |
-7.35 |
[-14.10; -0.59] |
0.034 |
Adjusted model 2 |
-8.76 |
[-15.88; -1.64] |
0.018 |
Reviewer 1-Round 2- Comment 2: The term "pilot study" should be introduced also in the title and/or in the abstract.
Answer 2: As requested, the term pilot study was added in the abstract.